# A Comparative Study of Ir(dmpq)₂(acac) Doped CBP, mCP, TAPC and TCTA for Phosphorescent OLEDs

**Despoina Tselekidou [1], Lazaros Panagiotidis [1], Kyparisis Papadopoulos [1], Vasileios Kyriazopoulos [2] and Maria Gioti [1,\*]**

1 Department of Physics, Nanotechnology Lab LTFN, Aristotle University of Thessaloniki, GR-54124 Thessaloniki, Greece
2 Organic Electronic Technologies P.C. (OET), 20th KM Thessaloniki-Tagarades, GR-57001 Thermi, Greece
\* Correspondence: mgiot@physics.auth.gr

**Abstract:** In this work, we present the fabrication and characterization of solution-processable red Phosphorescent Organic Light-Emitting Diodes (PhOLEDs). The proposed approach is based on Ir(III) complex, namely Bis(2-(3,5-dimethylphenyl)quinoline-C,N)(acetylacetonato)Iridium(III), also known as Ir(dmpq)₂(acac), which was doped in four different host materials: (a) 4,4′-Bis(N-carbazolyl)-1,1′-biphenyl (CBP), (b) 1,3-Bis(N-carbazolyl)benzene (mCP), (c) 1,1-Bis[(di-4-tolylamino)phenyl]cyclohexane (TAPC), and (d) tris(4-carbazoyl-9-ylphenyl)amine (TCTA). The metal–organic complex offers unique optical and electronic properties arising from the interplay between the inorganic metal and the organic material. The optical and photophysical properties of the produced thin films are investigated in detail using spectroscopic ellipsometry and photoluminescence, whereas the structural characteristics are examined by atomic force microscopy. This comparative study of the four different Host:Ir-complex systems provides valuable information to evaluate the emission characteristics in order to achieve pure red light. Finally, these materials were applied as a single-emissive layer in PhOLED devices, and the electroluminescence characteristics were studied.

**Keywords:** red PhOLEDs; doping; Ir(dmpq)₂(acac); spectroscopic ellipsometry; photoluminescence; electroluminescence; CBP; mCP; TAPC; TCTA

## 1. Introduction

Red organic light-emitting devices (OLED) containing phosphorescent emitters constitute an attractive research topic because of their various applications in different fields of medicine and energy [1–6]. It is well established that red OLEDs are of intense academic and industrial interest for energy-saving solid-state lighting as well as in flat-panel displays [3,4]. It is also important to mention that red OLEDs have also become promising light sources for compact and "imperceptible" biomedical devices that use light to probe, image, manipulate, or treat biological matter [1]. In this regard, the fabrication of red phosphorescent OLEDs (PhOLEDs) has attracted great attention.

The PhOLEDs are a breakthrough, as they theoretically exhibit their highest internal quantum efficiency in comparison to fluorescent devices. Especially, the maximum achievable internal quantum efficiency (IQE) was 25% for fluorescent devices emitting only a singlet exciton state. For this reason, special emphasis has been given to apply PhOLEDs as the strong spin-orbit coupling (SOC) and fast intersystem crossing could lead to a harvest of both singlet and triplet excitons in the emitting layer and achieve internal quantum efficiency as high as 100% theoretically [5–7]. The most promising strategy to fabricate PhOLEDs is based on the transition metal complex luminescent materials. Among the transition metal complexes, Ir (III) complexes are considered the most promising emitters in PhOLEDs based on their relatively high phosphorescence quantum yields, short triplet excited state lifetimes, excellent color tunability from blue to deep red, and splendid thermal and electrochemical stability [7–11].

It is notable that in PhOLEDs, Ir(III) complexes, as dopants, are implemented in host–guest systems, which enables us to influence the emission characteristics and the electroluminescence performance. The host material that forms the emission layer (EML) needs to satisfy a certain number of properties regarding the dopant. The host material with the phosphorescent emitter is very critical for achieving good performance in PhOLEDs with high color purity and selectivity. In particular, the highest occupied molecular orbital (HOMO) and the lowest unoccupied molecular orbital (LUMO) of the host and phosphorescent emitter dopant have to be carefully selected to achieve good exciton confinement within the EML [6,12]. Doping regulates the HOMO and LUMO of the emitting layer in addition to spin conversion of exciton from singlet to triplet [13–15]. This allows the energy bandgap to be adjusted to the desired level and thus generates light corresponding to that energy level. So, it has emerged as an efficient route to investigate the properties of Ir(III) complexes based on different host materials.

To date, these PhOLEDs are fabricated by sequential vapor deposition of several organic layers. This makes the deposition process, and consequently device fabrication, relatively difficult and expensive due to the high equipment cost, high energy consumption, and complicated fabrication process. For these issues, the vacuum deposition methods are unbeneficial for the large-scale low-cost industrialization of PhOLEDs. To overcome these problems, research efforts have been directed toward solution-processed Ir(III)-complexes. The wet-based technique is an attractive low-cost large area manufacturing tool because it facilitates easy co-doping of more than one dopant and provides compatibility with roll-to-roll manufacturing process [16,17]. It is therefore desirable to explore approaches for solution-processable materials based on the host–guest system.

Up to now, the most widely applied example of the solution-processable PhOLEDs is based on polymers as the host material for the emitting layers. More specifically, initial studies about the solution-processed PhOLEDs were focused on polymers as the host material, as it is well-known that polymers possess superior solubility in common solvents and good film-forming properties with low surface roughness. However, the intrinsic deficiencies of polymers' uncertain molecular structure, the requirement for a harsh purification method, and the low energy state limit the implementation of these materials in PhOLEDs. Another promising approach is based on small molecules as host materials. Compared to polymers, they possess easy synthesis, high purity, and stable thermal properties [18,19]. So, the combination of the synthesis of red emitters based on Ir(III)-complexes with different hosts and the simple wet-fabricated PhOLEDs remains an open issue in the research field of OLEDs.

In the present work, Bis(2-(3,5-dimethylphenyl)quinoline-C,N)(acetylacetonato)iridium(III), also known as Ir(dmpq)$_2$(acac), was doped in four different host materials such as 4,4′-Bis(N-carbazolyl)-1,1′-biphenyl (CBP), 1,3-Bis(N-carbazolyl)benzene (mCP), 1,1-Bis[(di-4-tolylamino) phenyl]cyclohexane (TAPC), and tris(4-carbazoyl-9-ylphenyl)amine (TCTA), and these Hosts:Ir-complex system were investigated as emitters in red solution-processable PhOLEDs. It is important to mention that these host materials possess good solubility in common solvents and film-forming properties, as it is a crucial factor in the solution fabrication process [14,15,17,20–22]. The metal–organic complex offers unique optical and electronic properties arising from the interplay between the inorganic metal and the organic material. For this reason, we mainly focus on the study of the optical and photophysical properties of these fabricated thin films not only for the Host:Ir-complex but also for the net host, which was carried out via spectroscopic ellipsometry and photoluminescence, respectively. According to this analysis, there is an opportunity to study if the energy transfer mechanism between the organic host material and the metal complex takes place. Additionally, since the film surface morphology of the EML layer is an important parameter that could be accounted for when introducing interfacial layers in the fabricated device, this work aims to give insight into the structural surface characteristics of the thin films using atomic force microscopy (AFM). Finally, these Hosts:Ir-complex were applied in solution-processable PhOLEDs as an emissive single-layer, and their electroluminescent properties were assessed. The determination of the optical properties in combination with the photo- and electro-emission characteristics

provide a thorough characterization and evaluation of photoactive materials and PhOLED devices in order to achieve high color purity and stability.

## 2. Materials and Methods

### 2.1. Ink Formulation

For the hole transport layer (HTL), a solution of poly-3,4-ethylene dioxythiophene: poly-styrene sulfonate (PEDOT:PSS, Clevios Heraus, Germany) AI 4083 mixed with ethanol in the ratio of 2:1 was prepared. For the host materials, 4,4′-Bis(N-carbazolyl)-1,1′-biphenyl (CBP), 1,3-Bis(N-carbazolyl)benzene (mCP), 1,1-Bis[(di-4-tolylamino) phenyl]cyclohexane (TAPC), and tris(4-carbazoyl-9-ylphenyl)amine (TCTA) were supplied by Ossila, Sheffield, UK. The Bis(2-(3,5-dimethylphenyl)quinoline-C,N)(acetylacetonato)iridium(III), also known as Ir(dmpq)$_2$(acac), was also supplied by Ossila Sheffield, UK. We prepared four blend materials with CBP, mCP, TAPC, TCTA as hosts and Ir(dmpq)$_2$(acac) as guest with a weight ratio of 96:4, respectively. These emissive materials were dissolved in toluene and stirred under heating for 24 h.

### 2.2. PhOLED Fabrication

The architecture structure of the solution-processed PhOLEDs and the molecular structure of the used phosphorescent dopant are illustrated in Scheme 1a,b, respectively. Firstly, pre-patterned Indium-Tin Oxide-coated glass substrates (received from Ossila Sheffield, UK) were extensively cleaned by sonication in DI, acetone, and ethanol for 10 min and followed by drying with nitrogen. Then, the substrates were transferred to the glove box, where the substrates were also treated with oxygen plasma at 40 W for 3 min. The PEDOT:PSS layer, which was used as the hole transport layer (HTL), was deposited by spin coating method onto the glass/ITO substrate and followed by annealing at 120 °C for 5 min. The emitting layers (EML) were spun at the same speed, specifically at a rotational speed of 2000 rpm for 1 min, onto the PEDOT:PSS layer. Finally, a bilayer of Ca 6 nm thick, and Ag 125 nm thick was used as a cathode electrode layer and was deposited using the appropriate shadow masks by vacuum thermal evaporation (VTE).

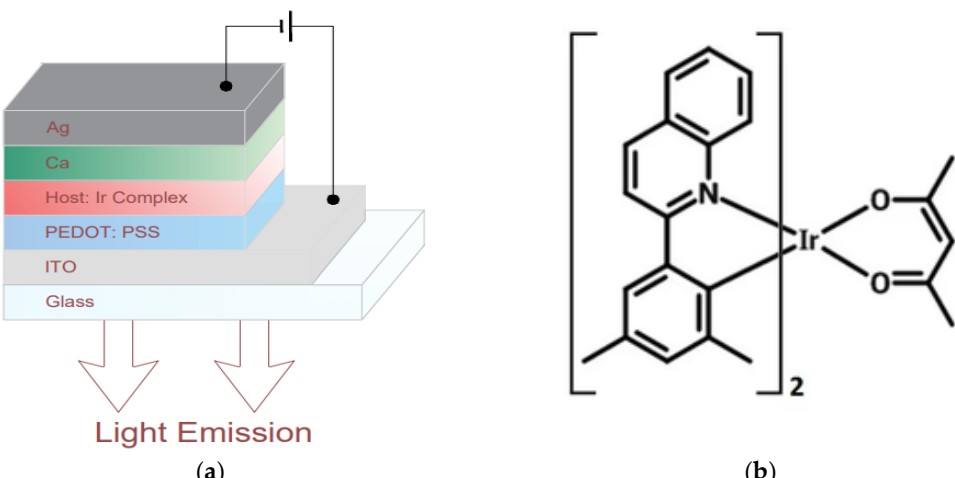

(**a**)                                      (**b**)

**Scheme 1.** (**a**) The architecture of the fabricated PhOLED devices, and (**b**) the molecular structure of the Ir complex.

### 2.3. Thin Film and Device Characterization

Spectroscopic ellipsometry (SE) is a powerful, robust, non-destructive, and surface-sensitive optical technique that allows the determination of the optical properties as well as the thickness of the light-emitting polymers. Through the SE technique, we can measure the pseudodielectric function $\langle\varepsilon(E)\rangle = \langle\varepsilon_1(E)\rangle + i\langle\varepsilon_2(E)\rangle$ of the studied thin films. In addition, by applying the appropriate modelling and fitting procedures we can extract significant information about the dielectric function $\varepsilon(\omega)$, the thickness of the thin

films with nanometer-scale precision, the absorption coefficient, and the optical constants, such as the fundamental band gap, and the higher energy optical gaps. The SE measurements were acquired using a phase modulated ellipsometer (Horiba Jobin Yvon, UVISEL, Palaiseau, France) at photon energies between 1.5–6.5 eV and 20 meV step at 70° angle of incidence. The SE experimental data were fitted to model-generated data using the Levenberg–Marquardt algorithm, taking into consideration all the fitting parameters of the applied model.

The surface morphology of the emitting thin films was investigated by atomic force microscopy (AFM) (NTEGRA, NT-MDT). The measurements were performed in ambient conditions, at tapping-mode operation, using silicon-based cantilevers with a high-accuracy conical tip and nominal tip roundness <10 nm.

Finally, the photoluminescence (PL) and electroluminescence (EL) characteristics of the active layers and the final PhOLED devices, respectively, were measured into the glove box, without encapsulation using the Hamamatsu absolute PL quantum yield measurement system (C9920-02) and the external quantum efficiency system (C9920-12), which measures brightness and light distribution of the devices.

## 3. Results and Discussion

### 3.1. Spectroscopic Ellipsometry

Spectroscopic ellipsometry (SE) in the NIR-Vis-fUV spectral region (1.5–6.5 eV) can provide valuable information on the optical and electronic properties, as well as the thickness of the metal–organic complexes films through the analysis of the measured pseudodielectric function $\langle \varepsilon(E) \rangle = \langle \varepsilon_1(E) \rangle + i\langle \varepsilon_2(E) \rangle$. At this point, it is important to mention that the optical properties are thus a function of both chemical nature and processing.

The investigation of the optical properties of pristine host materials CBP, mCP, TACT, and TCTA was initially realized in order to be used as a reference for the analysis of the $\langle \varepsilon(E) \rangle$ of the respective Host:Ir-complex. In order to extract quantitative information, we modelled and fitted the measured $\langle \varepsilon(E) \rangle$ spectra by applying a three-phase theoretical model which consists of the layer sequence air/host–material/glass according to the Levenberg–Marquardt minimization algorithm. The optical properties of the host materials were described using the modified Tauc–Lorentz (TL) oscillator model, which has been successfully applied in amorphous organic semiconductors [23]. More specifically, in the TL model, the imaginary part $\varepsilon_2(E)$ of the dielectric function is determined by multiplying the Tauc joint density of states by $\varepsilon_2(E)$ obtained from the Lorentz oscillator model and described by the following expressions [24]:

$$\begin{cases} \varepsilon_2(E) = \frac{1}{E} \cdot \frac{AE_0\Gamma(E-E_g)^2}{\left(E^2-E_0^2\right)^2+\Gamma^2E^2}, \ E > E_g \\ \varepsilon_2(E) = 0, \ E \leq E_g \end{cases} \tag{1}$$

The real part $\varepsilon_1(E)$ is obtained by the Kramer–Kronig integration as shown below [24]:

$$\varepsilon_1(E) = \varepsilon_\infty + \frac{2}{\pi} P \int_{E_g}^{\infty} \frac{\xi \varepsilon_2(\xi)}{\xi^2 - E^2} d\xi \tag{2}$$

where $\varepsilon_\infty$ is a constant that accounts for the existence of electronic transitions at higher energies, which are not taken into account in the $\varepsilon_2(E)$. The TL model provides the ability to determine the energy position of the fundamental band gap $E_g$, the amplitude A of the oscillator, the Lorentz resonant energy $E_0$, and its broadening term $\Gamma$. For each organic-host film, the appropriate number of TL oscillators was used for the accurate description of each individual dielectric response.

Concerning the Host:Ir-complex layers, we followed the same methodology of the above analysis in order to derive the optical and electronic properties. For the fitting analysis, the five-phase geometrical model air/Host:Ir-complex/PEDOT:PSS/ITO/glass was

applied, and the bulk dielectric function $\varepsilon(E)$ of the Host:Ir-complex films was described using the TL dispersion equation. The same number of TL oscillators was used for each Host:Ir complex as in the case of the respective pristine host material. This is justified by the fact that the percentage of the $Ir(dmpq)_2(acac)$ in the films was as low as 4%. Figure 1a–d show the calculated real $\varepsilon_1(E)$ and imaginary parts $\varepsilon_2(E)$ of the bulk dielectric function $\varepsilon(E)$ of the undoped and doped organic thin films, as they were reproduced using the best-fit parameters of the above analysis. Indeed, regarding the emissive layers based on Host:Ir-complex, it can be derived that their $\varepsilon(E)$ exhibit strong similarities with that of host materials. The comparison between the $\langle \varepsilon(E) \rangle$ of the measured (exp) and the theoretical (fit) ones of the doped films is demonstrated in the insets of the respective figures. The theoretical $\langle \varepsilon(E) \rangle$ spectra were derived based on the best-fit parameters of the analysis of the measured $\langle \varepsilon(E) \rangle$, which include the parameters of the applied dispersion equation and the thickness.

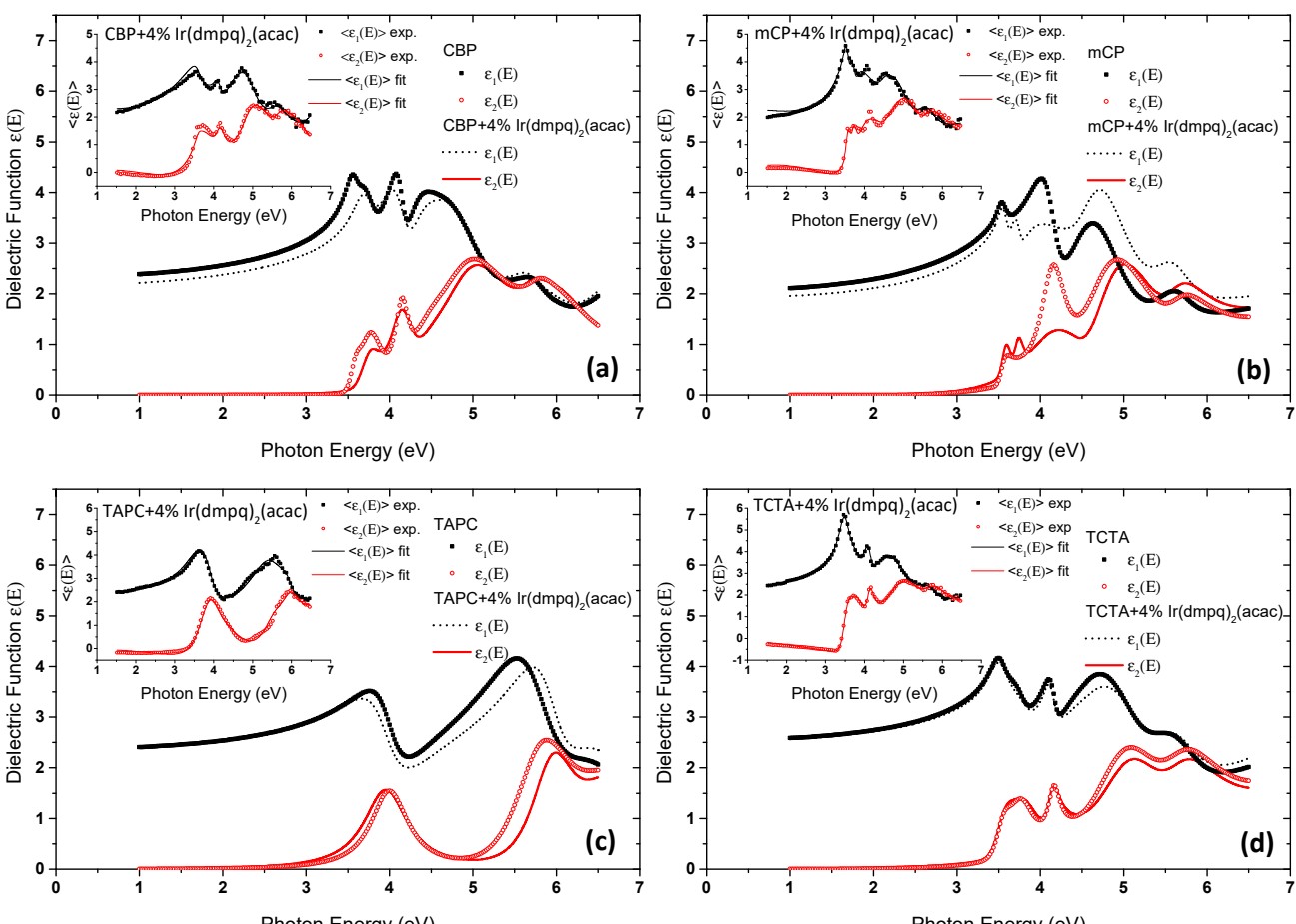

**Figure 1.** The bulk $\varepsilon(E)$ as they were calculated by the SE analysis of the undoped (symbols) and doped (lines) organic films: (**a**) CBP; (**b**) mCP, (**c**)TAPC; (**d**) TCTA. Insets show the experimental (symbols) $\langle\varepsilon(E)\rangle$ and the corresponding fitted ones (lines) of the studied doped organic films.

It can be easily recognized that all studied hosts exhibit low electronic absorption up to 3 eV or above. Their doping with the $Ir(dmpq)_2(acac)$ affects mainly the characteristics of the absorption edge as it is can be easily deduced by the calculated absorption coefficients of the films. These results are illustrated in Figure 2a–d. In all films, an increase of the absorption in the sub-bandgap range was obtained, and the most pronounced are that of CBP and mCP. It should be noted here that there are only small modifications in the characteristics of the electronic absorptions of all the host organic materials at higher energies.

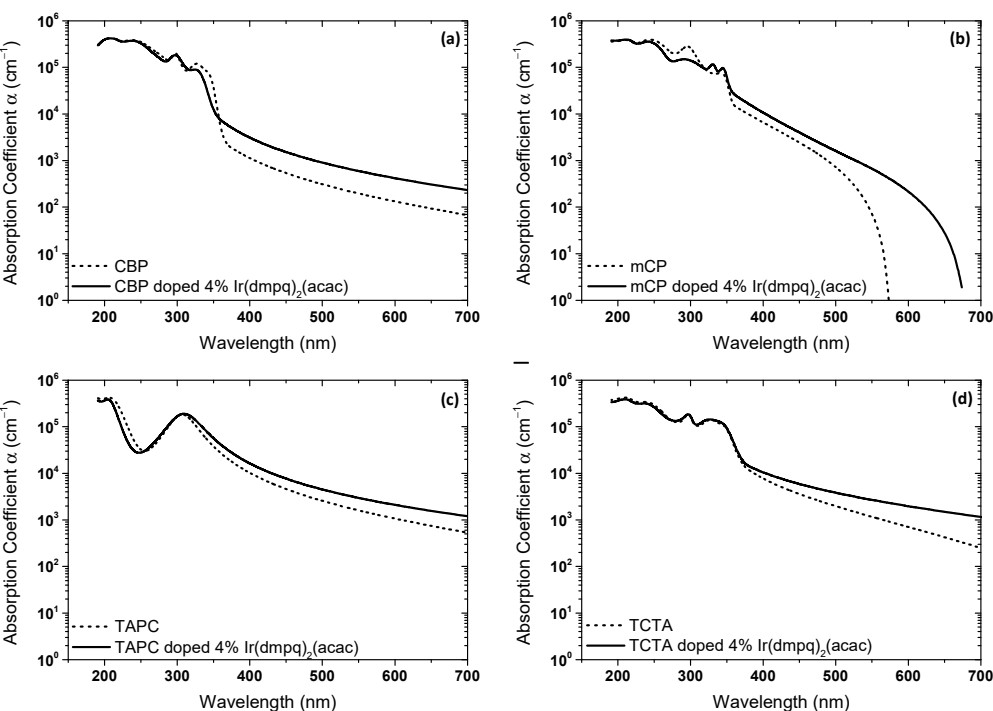

**Figure 2.** The calculated absorption coefficients, by the SE analysis, versus wavelength of the undoped (dashed lines) and doped (solid lines) organic films: (**a**) CBP; (**b**) mCP; (**c**)TAPC; (**d**) TCTA.

There are several absorption bands for each material. In principle, the absorption coefficient spectrum of Ir(III)-compound doped in CBP host exhibits a similar absorption band with the absorption spectrum of net CBP. In the CBP spectrum, the absorption peak around 295 nm could be assigned to the carbazole-centered π–π * transitions, whereas the absorptions in the range of 319–340 nm could be attributed to π–π * transitions between the carbazole unit and the central biphenyl unit in the molecule [13,22,25,26]. On the other hand, for Ir(III)-complex doped in CBP host, the absorption between 250–350 nm is CBP-host and ligand-centered (LC) based. More specifically for the LC transition, we assign the strong absorption band in the UV region to the spin-allowed π–π * transition of the cyclometalated quinoline ligands of the Ir(III)-compound. The weak absorption band at wavelengths higher than 350 nm could be assigned to a handful of charge-transfer transitions. In particular, this could be assigned to the admixed MLCT (metal-ligand charge transfer) and MLCT/LC transitions, the latter of which are usually spin-forbidden but were allowed due to the strong spin-orbital-coupling (SOC) induced by the heavy metal-Iridium(III) [2,7,9,12,27–29].

Furthermore, comparing the metal–organic Ir(III)-complex based on mCP host with the net mCP material, they present similar spectral features. The absorption spectra of the net mCP exhibit three absorption bands located at 250–350 nm, which could be associated with the n–π * and π–π * transitions of the carbazolyl units [30,31]. It is worth noting that the doping of the mCP with the organometallic complex brings about some differences in the absorption edge, and this may assign to the states within the energy gap of the host material. It is recognized that the absorption tail of the Ir(III)-complex doped in the mCP is higher, in the range from 360 nm and above, and this can also be associated with the MLCT transitions.

Moreover, both TAPC and Ir(III)-compound doped in TAPC exhibit similar absorption characteristics. However, the broadband absorption located at 313 nm is evident in the case of the Ir(III)-complex based on TAPC, and a noticeable absorption tail was also observed. It is also significant to refer that we observe differences between the absorption features based on TAPC and the other host–organic materials absorption spectrum. This may be attributed to the fact that TAPC is composed of two tri(p-tolyl) amine (TTA) molecules

chemically bridged by a cyclohexane ring and present different absorption characteristics with the host materials based on the carbazolyl compound [32,33].

The TCTA and Ir(III)-complex doped in TCTA present a similar shape of absorption. In particular, the absorption peaks at 293 and 327 nm could be assigned to n–π* and π−π* transition of triphenyl amine and carbazole, respectively. In the absorption spectrum of metal–organic complex doped in TCTA, the absorption tail is obvious in the range from 370 nm and above, and this can be attributed to the MLCT/LC transitions [34,35].

### 3.2. Photoluminescence

The PL emission spectra of the Host:Ir-complex films are illustrated in Figure 3. In the host–guest system, such as the Host:Ir-complex, the energy transfer between the host and the guest is the main emission mechanism. Generally, the energy transfer is that the excitons are primarily formed in the host and then transfer their energy to the guest through the Förster and/or Dexter mechanisms [36]. In order to clarify the emission mechanism of Host:Ir-complex we also measured the PL spectra of host materials, which are plotted in the same figure, for completeness. The PL spectra were recorded upon excitation at 340 nm. It is obvious for all Hosts:Ir-complex the dominant PL emission band is located in the region between 560–800 nm. On the other hand, the PL emission profile of CBP, mCP, and TCTA host materials is centered at 350–500 nm, in the blue region, whereas that of TAPC covers a significantly wider range up to 700 nm. As can be seen, the emission from the hosts is quenched by Ir(dmpq)$_2$(acac). This fact indicates that the efficient energy transfer mechanism from the host to the guest takes place [37]. However, it can be seen that a negligible emission has existed in the region between 380–480 nm, which is assigned to the emission of host materials. In addition, the broad, structureless spectral features lead us to conclude that the phosphorescence originates primarily from the MLCT states [7,10,28]. Particularly, the dominant PL peak for all Hosts:Ir-complex was centered at 620 nm and there is a subtle shoulder peak at approximately 650 nm except for the Ir(III) complex doped in TAPC. The latter Host:Ir-complex exhibits PL emission maximum at 612 nm, and compared to the other three, a blue shift is observed. This fact could be associated with the PL emission of host-TAPC.

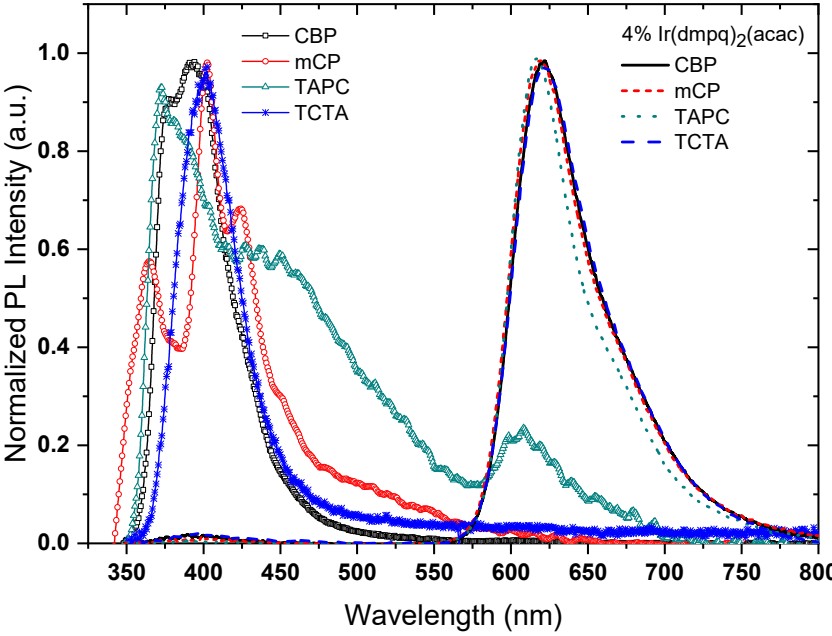

**Figure 3.** The normalized PL spectra of the Host:Ir-complex films in comparison to the net hosts.

### 3.3. Atomic Force Microscopy

Figure 4a–d present the AFM height images of spin-coated thin films in order to evaluate the film-forming ability, morphological properties of the mixed films, and the effects of dopant distribution on film morphology. Moreover, the AFM results, as listed in Table 1, show the values of the mean roughness (Sq), the root mean square (Sa), the peak to peak (Sy), and the calculated, through the SE analysis, thicknesses of the Hosts:Ir-complex films. One can observe that the surface morphology of all samples was homogeneous and adequately covers the substrate. The image analysis revealed that smooth and continuous films were formed with low Root Mean Square roughness (RMS) values. In more detail, by comparing the RMS values of the Host:Ir-complex films, it is found that doping of Ir(dmpq)$_2$(acac) results in the formation of smooth thin films, as the RMS values are below 0.32 nm for all Hosts:Ir-complex. The thin films are continuous with smooth surface morphology and quite small RMS values, which means that the Host:Ir-complex exhibits morphological stability without any obvious particle aggregation or phase separation. All these desirable features are favorable for host-dopant combination to be used in PhOLED fabrication and operation processes.

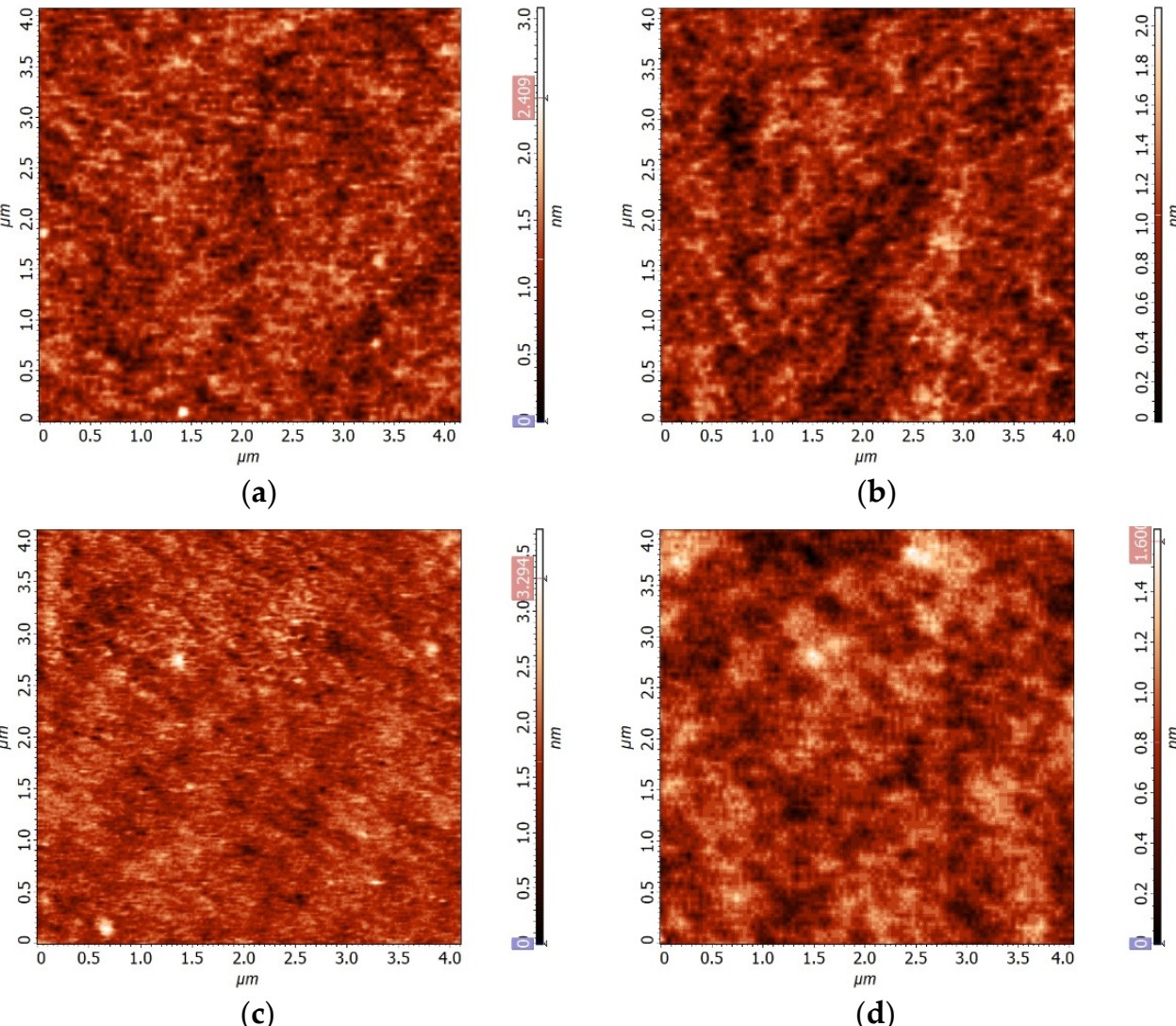

**Figure 4.** AFM topography images scan size 4 × 4 µm of the Ir(III)-complex doped in: (**a**) CBP; (**b**) mCP; (**c**) TAPC; (**d**) TCTA.

**Table 1.** AFM and SE results for the spin-coated thin films.

| Ir(dmpq)$_2$(acac) Doped in | Thickness by SE (nm) | Root Mean Square Sq (nm) | Average Roughness Sa (nm) | Peak to Peak Sy (nm) |
|---|---|---|---|---|
| CBP | 73 | 0.30 | 0.24 | 3.07 |
| mCP | 52 | 0.28 | 0.22 | 2.08 |
| TAPC | 43 | 0.32 | 0.25 | 3.73 |
| TCTA | 43 | 0.23 | 0.17 | 1.65 |

*3.4. Electroluminescence*

We have investigated the potential of these Hosts:Ir-complex as emissive materials in phosphorescent OLED applications using devices having the configuration glass/ITO/ PEDOT:PSS/Host:Ir-complex/Ca/Ag. Figure 5a–d show the respective experimental EL spectra of the studied devices, which were obtained at 12 V, as well as the corresponding PL spectra for comparison. For their better evaluation, a deconvolution fitting analysis of the experimental EL and PL spectra was realized, revealing the existence of three main peaks. The results of this analysis were the wavelengths where the maximum of the emission of the films is located and the full width at half maximum (FWHM). According to the EL deconvolution analysis, it is noteworthy to mention that all EL spectra from the Ir(dmpq)$_2$(acac) doped in different hosts exhibit emission approximately between 550–750 nm. The EL emission of the host is completely quenched, and the dopant emission completely dominates and results in a red emission from the Ir(dmpq)$_2$(acac).

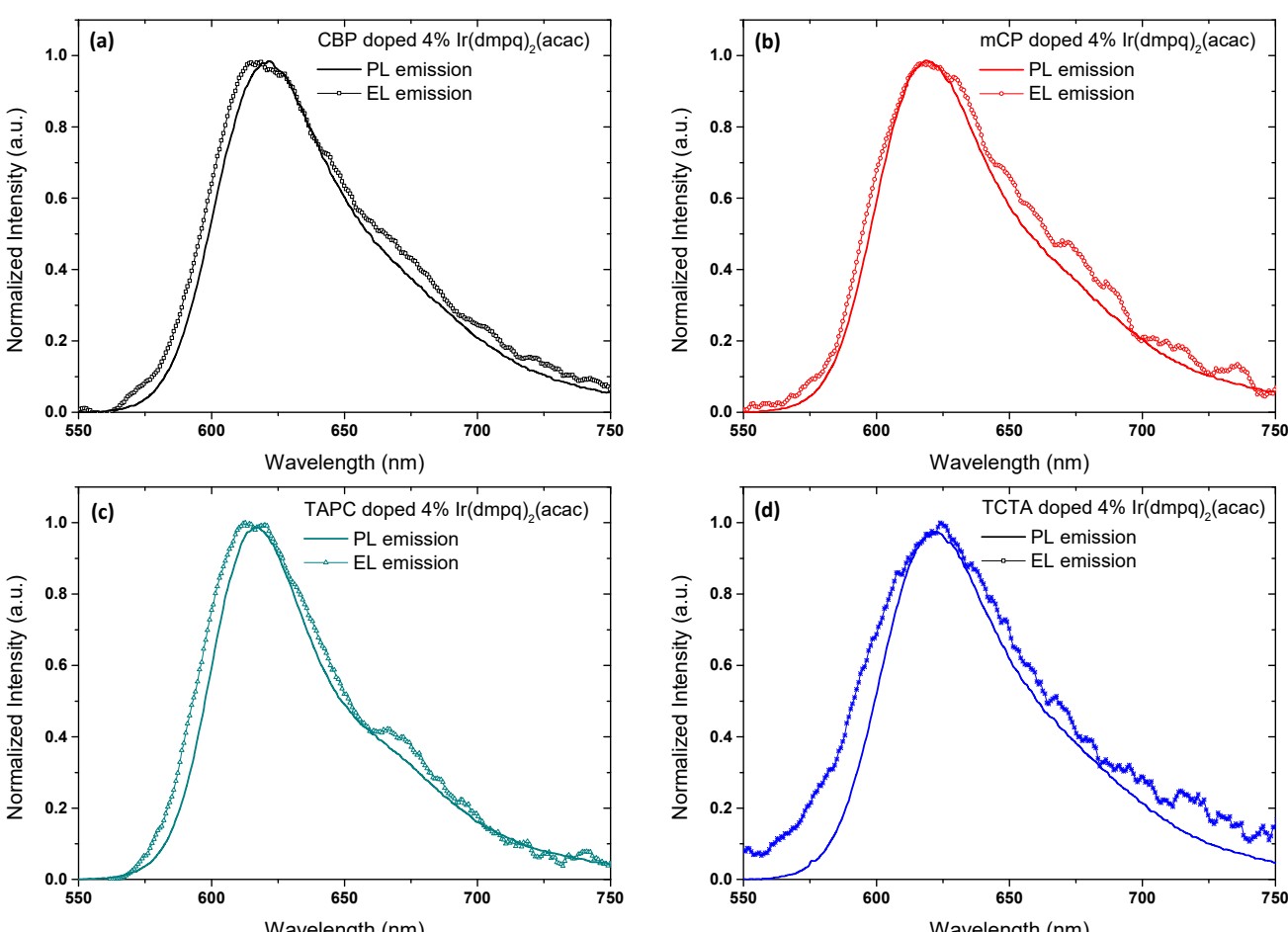

**Figure 5.** The normalized EL spectra of the PhOLED devices of the Ir-complex doped in: (**a**) CBP; (**b**) mCP; (**c**) TAPC; (**d**) TCTA as well as the respective PL spectra for comparison.

Compared to the PL and EL emission, it is obvious that the tendency of the EL spectra was similar to the PL spectra. However, for all studied Hosts:Ir-complex, the EL emission profile is blue-shifted in comparison to the PL emission profile. According to the deconvolution analysis, there are fundamental differences in the shifts of the peaks either between the PL and EL of the different peaks of the same Host:Ir-complex film or between the same peak of the different films. These results are demonstrated in Figure 6a, in which the horizontal lines indicate the mean wavelength values of the three peaks, and the arrows denote the shift between the respective PL and EL peaks. Thus, we can distinguish that the smaller peak shifts are obtained for the Ir(dmpq)$_2$(acac) doped CBP and the larger for the Ir(dmpq)$_2$(acac) doped TCTA.

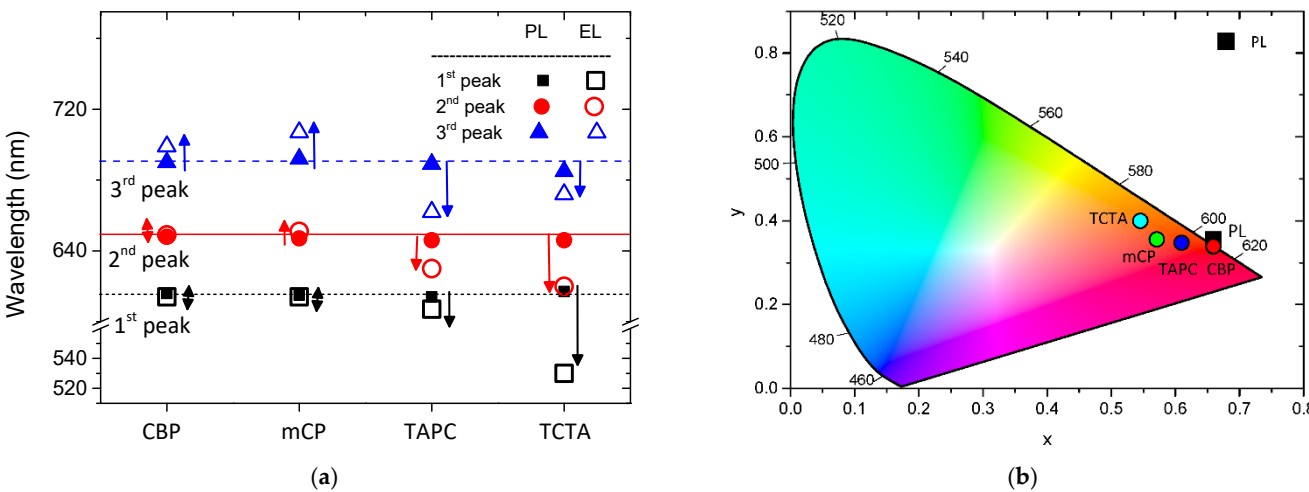

(a)                                                                                      (b)

**Figure 6.** (**a**) Schematical representation of the $\lambda_{max}$ of the peaks calculated through the deconvolution analysis of the PL and EL emission; (**b**) the CIE color coordinates relative to the PL and EL spectra of the four corresponding Host:Ir-complex.

It is remarkable to refer that the PL and EL emission is based on different mechanisms. It is established that in PL, the emission results from the radiative recombination of the photo-excites carriers. On the other hand, the EL depends not only on the optical properties and physical properties of the emitting layers but also on the electrical properties of two conductive regions which are used for injection and transportation of carriers [38]. So, the EL emission is related to the two mechanisms, energy transfer and charge trapping. As it has been already mentioned, in host-dopant systems, three paths can lead to phosphorescent emission by the dopant. It has been suggested that (i) in the host molecule, the generated singlet excitons under electrical excitation can be transferred to the singlet excited state of the dopant via Förster and Dexter energy transfer processes. Then, they can be converted into triplet excitons for radiative decay by an efficient intersystem crossing (ISC) process. (ii) The other possible scenario for the host molecule is based on generated triplet excitons, which can be transferred to the triplet excited state of a phosphorescent dopant through Dexter energy transfer. (iii) The holes and electrons can directly recombine in the phosphorescent dopant by charge trapping mechanism [39]. We speculate that the emission in the red region could be assigned to both the complete energy transfer mechanism from the host matrix to the guest-Ir(dmpq)$_2$(acac) and charge trapping in Ir(dmpq)$_2$(acac). According to our results, we can assume that excitons on host molecules, triplet or singlet, are formed by the capture of opposite charges that are injected from both electrodes. These excitons then transfer their energy to the nearby guest molecules through the Förster mechanism or Dexter mechanism. At this point, it is also important to mention that the direct trapping of charges on Ir(III)-compound followed by guest exciton formation and radiative decay may be another mechanism responsible for the guest emission [37,40,41].

The chromaticity diagram with the Commission Internationale de L' Eclairage (CIE) coordinates, which are derived from PL and EL measurements, is illustrated in Figure 6b.

The CIE coordinates of Hosts:Ir-complex, confirm the PL emission in the red region. It was found that the emission of Hosts:Ir-complex is located at the edge of the red region in the CIE coordinates map. This red emission from Host:Ir-complex can lead to the assumption that the efficient energy transfer takes place from the singlet-excited state in the host to the (Singlet) 1MLCT band of the guest, Ir(dmpq)$_2$(acac), followed by fast intersystem crossing to the triplet state 3MLCT of Ir(dmpq)$_2$(acac) and, consequently, emission from its triplet state.

In the case of EL emission, it can be observed that the devices based on Host:Ir-complex emitted reddish–orange light. In more detail, the CIE coordinates are (0.660, 0.339) for the device based on the Ir compound doped in CBP. Note that these coordinates are very close to the National Television System Committee (NTSC) standard for red subpixels (0.67, 0.33) [42]. The device of Ir-complex doped in TAPC exhibit CIE coordinates very close to the ideal red emission, the values are (0.610, 0.347). In the case of the other two Hosts:Ir-complex, the CIE coordinates shifted to the orange emission. Specifically, Ir-complex doped in mCP and TCTA obtain values at (0.571, 0.356) and (0.546, 0.400), respectively. Thus, from the comparison of the EL and PL emission spectra of each emitting material, the Ir(III)-complex doped in CBP exhibits the red color selectivity in its emission both in a thin film form and in a PhOLED device.

Finally, the performances of the fabricated PhOLED devices were evaluated by measuring their current density–voltage (J–V) and luminance–voltage, and the results are depicted in Figure 7a,b, respectively. The luminance measured for the Ir compound doped in CBP reaches 293 cd/m$^2$ at 14 V, whereas when the Ir compound doped in the other three hosts mCP, TAPC, and TCTA present lower luminance values. Specifically, the luminance measured for the Ir compound doped in mCP is 91 cd/m$^2$ at 14 V, for the Ir compound doped in TAPC is 62 cd/m$^2$ at 10 V and for the Ir compound doped in TCTA is 45 cd/m$^2$ at 14 V. The comparison of the highest luminance values between the photoactive films can be justified by their thicknesses, listed in Table 1.

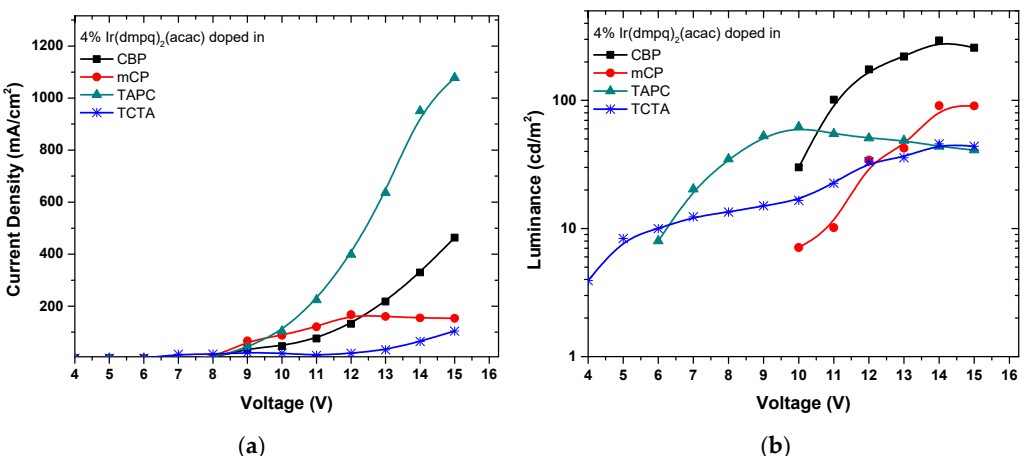

**Figure 7.** (**a**) The current density–voltage (J–V) characteristic curves and (**b**) logarithmic plot of luminance–voltage of the fabricated PhOLED devices.

To enforce the red color purity improvement, the emissive layers based on the Host:Ir complex, are newly proposed in this study. Thus, in this work, we compare the Ir(III) complex doped in four different host materials. Among the Hosts:Ir complex, the Ir(dmpq)$_2$(acac) organometallic compound doped in CBP is the most promising material to achieve red emission stability and selectivity. So, the Ir(dmpq)$_2$(acac) doped in CBP is an encouraging step forward to achieve red solution-processable PhOLEDs, as it provides good exciton confinement within this emitting layer and the CIE coordinates obtained from the EL measurements approach the ideal red-light emission. A thorough investigation of the fabrication parameters and devices' architectures could be considered a decisive factor for improving the efficiency of the produced PhOLED devices.

## 4. Conclusions

In conclusion, we have studied the Ir-complex doped in four different host materials, namely CBP, mCP, TAPC, and TCTA for solution-processed red PHOLEDs. A thorough investigation of the absorption and emission behavior of the Ir(III)-complex doped in these host materials is presented. Afterward, these materials were applied as a single-emissive layer in the wet fabrication of PhOLED devices using the spin-coating process. A comprehensive study of the PL and EL emission of the spin-coated thin films was also realized in order to evaluate the color selectivity. We found that the devices based on Host:Ir-complex demonstrate emission in the orange–red region. Compared to the host materials, Ir(III) complex doped in CBP is a promising candidate material in order to achieve red-light emission from the phosphorescent device, as the CIE coordinates are very close to the National Television System Committee (NTSC) standard for red subpixels.

**Author Contributions:** Writing—original draft preparation, investigation, formal analysis, data curation, D.T.; methodology, investigation, writing—review and editing, data curation, L.P.; investigation, methodology, validation, writing—review and editing, K.P.; data curation, validation, writing—review and editing, V.K.; supervision, conceptualization, visualization, writing—review and editing, funding acquisition, M.G. All authors have read and agreed to the published version of the manuscript.

**Funding:** This research has been co-funded by the European Regional Development Fund of the European Union and Greek national funds through the Operational Program Competitiveness, Entrepreneurship, and Innovation, under the call RESEARCH—CREATE—INNOVATE (project code: T1EDK-01039).

**Institutional Review Board Statement:** Not applicable.

**Informed Consent Statement:** Not applicable.

**Data Availability Statement:** Data presented in this article is available on request from the corresponding author.

**Conflicts of Interest:** The authors declare no conflict of interest. The funders had no role in the design of the study; in the collection, analyses, or interpretation of data; in the writing of the manuscript, or in the decision to publish the results.

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
