# Peer review of "A Comparative Study of Ir(dmpq)2(acac) Doped CBP, mCP, TAPC and TCTA for Phosphorescent OLEDs"

_photonics, doi:10.3390/photonics9110800_

Round 1

Reviewer 1 Report

The presented work is devoted to the fabrication and characterization of solution-processable red Phosphorescent Organic Light-Emitting Diodes based on Ir(III) complex (Ir(dmpq)2(acac)), which was doped in four different host materials: CBP, mCP, TAPC and TCTA. Research in this direction is relevant and of great interest for the development of organic electronics and photonics. From reading the article, I got a good impression: the need for research in the introduction is well and thoroughly justified, the materials are well equipped with experimental and theoretical studies, and the presented results are well described and illustrated. In accordance with the above, I recommend this work for publication in the Photonics.

Based on the presented results of the AFM study, I would like to recommend the authors to pay attention to the following in the future. For more accurate analysis of dielectric surfaces and determination of their parameters using the AFM method, before registration, it makes sense to carry out the procedure for removing the electrostatic charge (increasing humidity inside the device chamber, grounding the sample surface), since it can greatly affect the result [https://doi.org/10.1134/S1027451008050108].

Reviewer 2 Report

  In a manuscript with the ID photonics-1920677, entitled “A comparative study of Ir(dmpq)2(acac) doped CBP, mCP, TΑPC and TCTA for phosphorescence OLEDs” the authors describe four common host materials and also the commercially available iridium complex. Spincoated films of the hosts and also their blends with the guest complex are characterized with ellipsometry and their dielectric functions together with absorption spectra are derived from a theoretical model and photoluminescence of the samples is measured. Subsequently, the topographies of the blended films are registered with atomic force microscope and finally simple organic light-emitting diodes with the doped four hosts are build and their electroluminescence (EL) is measured. Based on the EL spectra the Commission Internationale de l'Eclairage (CIE) coordinates are calculated for all the devices and the purest red-emitting OLED is determined.

Detailed comments

Line 2: In the name of the complex “2” should be as a subscript.

Line 3: “phosphorescent” instead of “phosphorescence”.

Line 36: “(PhOLEDs)” should be here instead of line 43.

Line 50: “electroluminescence”, not “Electroluminescence”.

Line 57: change “host layer” to “emitting layer”.

Line 110: remove “3. Results”.

Line 118: superscript in 120oC.

Line 118: “The emitting layers (EML) were spun at the same speed onto the PEDOT:PSS layer”. No information is given about the speed, please specify.

Line 123: Add molecular structures in the figure, especially of Ir(dmpq)2(acac). There are similar complexes with methyl groups attached in different positions, the molecular design will dispel doubts about the molecular structure.

Line 141: Were devices encapsulated in inert atmosphere? Calcium is very sensitive to oxygen.

Line 185: “it can be derived that their”

Line 226: “doping of the mCP with the organometallic complex” instead of “doping of the mCP in the organometallic complex”

Line 242: “the absorption tail is obvious in the range from 370 nm and above, and this can be attributed to the MLCT transitions [33,34].” Absorption spectrum of the complex in Adv. Mater. 2011, 23, 2721, DOI: 10.1002/adma.201100405 has π-π* transitions between 400 – 450 nm.

Line 266: “which may be ascribed to ligand-centered emission, demonstrating the effective mixing of the MLCT and interligand levels [2].” In reference 2, two iridium complexes different from Ir(dmpq)2(acac) are compared therefore similar spectral features in the reference and in the manuscript should not be assigned to the same explanation.

Line 279: “it is found that doping of Ir(dmpq)2(acac) does not affect the morphology of thin films” It is not possible to determine that since AFM topography for undoped samples is not presented.

Line 331: “molecules” instead of “chains”

General remarks

All the materials investigated by the authors have been studied since many years and most of the results presented in the manuscript can be found in earlier publications.

Ellipsometry measurements done by others:

CBP: Z.T. Liu et al. / Synthetic Metals 150 (2005) 159, doi: 10.1016/j.synthmet.2005.02.001

              CBP and mCP: Chem. Mater. 2013, 25, 3038, DOI: 10.1021/cm4011597

              TAPC: Adv. Mater. 2012, 24, 6368, DOI: 10.1002/adma.201202422

              PHYSICAL REVIEW LETTERS 120, 017402 (2018), DOI: 10.1103/PhysRevLett.120.017402

              TAPC and TCTA: Organic Electronics 88 (2021) 106014, DOI: 10.1016/j.orgel.2020.106014

It is true that in the manuscript the measurements are up to 6.5 eV while the results of other authors do not arrive to such high energy, however for applicative purpose the visible range (< 3.26 eV or > 380 nm) is crucial. The sample preparation method (evaporation vs spincoating) should have minor effect on the results because the molecules create isotropic layers as present in most of the above publications where ordinary and extraordinary refractive indices are similar. Also, the PL properties of layers prepared in the two ways should be very similar (see J. Phys. Chem. C 2020, 124, 11701, DOI: 10.1021/acs.jpcc.0c01968). Finally, doping the hosts with only 4% of the iridium complex should not affect much the optical constants of the samples. In figure 1 (b) in the manuscript big difference can be noticed between the doped and undoped films but the authors do not comment this.

PL measurements done by others:

              CBP: J. Phys. Chem. C 2020, 124, 11701, DOI: 10.1021/acs.jpcc.0c01968

              mCP: SCIENTIFIC REPORTS 4 : 7009,  DOI: 10.1038/srep07009

              TAPC: ACS Appl. Mater. Interfaces 2017, 9, 10955, DOI: 10.1021/acsami.6b16094

              Chemical Physics 256 (2000) 351, DOI: 10.1016/S0301-0104(00)00123-3

              TCTA: Micromachines 2022, 13, 51, DOI: 10.3390/mi13010051

Absorption spectra of some of the molecules can also be found in the articles indicated above.

Authors of the manuscript calculate absorption coefficients (Fig. 2) but absorption measurement is probably the most common spectroscopical technique which can give the exact spectrum derived from an experiment while simulations can be less trustful. In fact, in figure 2 a, c, d in the manuscript the spectra have tails till 700 nm, while in figure 2 b the absorption coefficient decreases rapidly before that wavelength. Taking into consideration the absorption spectrum of the complex in Adv. Mater. 2011, 23, 2721, DOI: 10.1002/adma.201100405, where the complex does not absorb above 650 nm, the last figure seems to be correct and figures 2 a, c, d not. On the other hand, small doping level of the samples could have little effect on their absorption. All these doubts can be easily resolved with absorption spectra measured experimentally.

In line 86 the authors write “According to this analysis, the emphasis will be placed on the energy transfer mechanism between the organic host material and the metal complex”. In the manuscript there is a discussion of energy transfer mechanisms in organic semiconductors but only at a general level. There is no analysis of the overlap of the complex absorption spectrum with the PL spectra of the hosts, no Forster radii are calculated or samples with different doping levels are not investigated, no transient PLs or lifetimes are measured. From the analysis done by the authors no information can be derived which host transfers energy to the dopant the best and why.

EL measurements with the same complex done by others:

              Adv. Mater. 2011, 23, 2721, DOI: 10.1002/adma.201100405

              Adv. Funct. Mater. 2013, 23, 4105, DOI: 10.1002/adfm.201300187

              ACS Photonics 2015, 2, 9, 1366, DOI: 10.1021/acsphotonics.5b00346

              NATURE COMMUNICATIONS 5:4769, DOI: 10.1038/ncomms5769

In all the above articles OLEDs were evaporated while in the manuscript the authors produce organic layers with spincoating. However, a device production method should not have strong influence on the shape of EL spectrum (the emitter is the same).

Moreover, all the host molecules investigated by the authors are predominantly hole transporting materials and in order to achieve balanced charge transport they are typically mixed with electron transporting compounds in spincoated emitting layers. Alternatively, electron transporting layer should be placed on the top of emitting layer to obtain recombination zone far from the electrode, which allows to arrive at higher efficiencies (the performance of the OLEDs is not reported by the authors).

In summary, it is not clear how some of the measurements help to achieve the goal of the investigation, which seems to be pure red EL. The ellipsometry analysis of dielectric functions has no conclusions and it is never mentioned in other sections of the manuscript. The absorption and PL spectra are discussed in more details but these results are known from earlier publications. The energy transfer is described very generally without entering into details of the processes of the particular host:complex pair. Finally, from the comparison of the PL and EL spectra, the shifts of the peaks between the two cases are reported but a discussion of the origins of those shifts is missing.

   Taking into account a small novelty of the data and all the above flaws I do not recommend the article to publication in Photonics.

Reviewer 3 Report

The authors investigated the optical and photophysical properties of films of a red-emitting Ir complex Ir(dmpq)2(acac) in four different hosts fabricated by solution-process, which are quite interesting. However, the EL properties were not sufficiently studied. Therefore, I suggest the authors improve the EL section.

1. Luminance-current-voltage and efficiency-luminance characteristics are essential for OLEDs. In addition to the EL spectra, the authors should also discuss other EL properties of Ir(dmpq)2(acac) in different hosts.

2. Generally, the optimum doping concentration for phosphorescent emitters is around 10 wt%. I wondered why the authors chose the concentration of  4 wt% in all samples. The results of different concentrations should be provided.

Round 2

Reviewer 2 Report

  The manuscript has been improved and new data about electroluminescence performance have been added, but there are still two main flaws.

  In the introduction, a paragraph about the importance of molecules wet deposition with respect to more commonly used polymers should be included. The authors’ answers to the previous review contain some good points suitable for this discussion, they can be used in the introduction together with proper references. The inclusion of such a discussion will help to understand better a reader why the investigation presented by the authors is important.

  The main flaw, in my opinion, is a lack of the connection between characterizations done by the authors. The reason why exactly those experiments were done to characterize the materials and not some others is elusive. The cause of that is there is no hypothesis/question that the authors try to confirm/answer in the manuscript. It is true that the title says “A comparative study of …” which suggests such a type of presentation of data, but as was pointed out in the previous review, many of experimental data reported by authors for the investigated materials have been already published. The manuscript would improve significantly if the authors could find some links between the experiments. For example, the SE measurements can give information about the microstructure of films (the claim taken from the authors’ answer in point 17). Could the AFM measurements confirm microstructure information derived from the SE measurement? Can the information about the film structure be connected with the PL and EL spectra?

As an example of what type of reasoning I mean, the authors can see J. Phys. Chem. C 2020, 124, 21, 11701, DOI: 10.1021/acs.jpcc.0c01968. In that article, the measurements take their origin from the questions that raised from the data of the previous experiment or are done to prove the hypothesis.

The question the authors may try to answer in their manuscript is why CBP turned out to be the best host for the Ir complex. The response can give some indications for hosts used in the future that will allow to achieve improved properties of other host : Ir complex systems.

  In summary, I believe the manuscript can be improved and recommend it to a major revision.

Additional remarks to the improved manuscript:

Line 39: to apply

Line 87: “A fundamental understanding of their basic properties is a significant step forward in the development of red phosphorescent materials.”

What indications for the development of red phosphorescent materials can be derived from the experiments described in the manuscript?

Line 109: Is it a molar, weight or volume ratio? Please specify.

Line 387: “To specify the energy transfer mechanism from the host material to the Ir-compound, the optical and photophysical characterization of host materials was also carried out.”

This phrase should be removed. The optical and photophysical characterization of the host materials did not specify the energy transfer mechanism. The authors discuss energy transfer mechanisms and assign the best explanation for the used host:guest system, but that is not based on the results of the experiments. For example, I could claim that the Dexter energy transfer is involved, but optical and photophysical characterization presented by the authors can’t confirm or deny that. We know that this mechanism is not involved, because in fluorescent hosts the Forster transfer is dominant, but this information can’t be derived from the presented measurements. Similar discussion can be done for the sentence in line 393. The results do not prove what the authors claim, a general knowledge about such organic systems does it.

Remarks to the authors’ answers to the review 1

Point 15: Line 279: “it is found that doping of Ir(dmpq)2(acac) does not affect the morphology of thin films”. It is not possible to determine that since AFM topography for undoped samples is not presented.

Response 15: We show only the AFM images of the topography of the doped materials, as these materials were applied, as emitting layers in PhOLED devices. The importance of this topic is related to the fact that the interfacial properties between the thin films sets the condition for the injection of charges in a device. Generally, it was established that the smoother surface reduced the loss of the injection and transportation of charges at the interface and finally confined the exciton within the emissive layer, which may be beneficial in OLED devices consisting of different layers. So, in this work, the AFM images are evidence of the smooth formation of thin films.

I agree with the authors’ response, however my remark about line 279 is still valid. It is necessary to rephrase the statement “…does not affect the morphology of thin films, …” into something like “… results in the smooth formation of thin films”.

Point 17: All the materials investigated by the authors have been studied since many years and most of the results presented in the manuscript can be found in earlier publications. Ellipsometry measurements done by others:

 CBP: Z.T. Liu et al. / Synthetic Metals 150 (2005) 159, doi: 10.1016/j.synthmet.2005.02.001

CBP and mCP: Chem. Mater. 2013, 25, 3038, DOI: 10.1021/cm4011597

TAPC: Adv. Mater. 2012, 24, 6368, DOI: 10.1002/adma.201202422 PHYSICAL REVIEW LETTERS 120, 017402 (2018), DOI: 10.1103/PhysRevLett.120.017402

TAPC and TCTA: Organic Electronics 88 (2021) 106014, DOI: 10.1016/j.orgel.2020.106014

 It is true that in the manuscript the measurements are up to 6.5 eV while the results of other authors do not arrive to such high energy, however for applicative purpose the visible range (< 3.26 eV or > 380 nm) is crucial. The sample preparation method (evaporation vs spincoating) should have minor effect on the results because the molecules create isotropic layers as present in most of the above publications where ordinary and extraordinary refractive indices are similar. Also, the PL properties of layers prepared in the two ways should be very similar (see J. Phys. Chem. C 2020, 124, 11701, DOI: 10.1021/acs.jpcc.0c01968). Finally, doping the hosts with only 4% of the iridium complex should not affect much the optical constants of the samples. In figure 1 (b) in the manuscript big difference can be noticed between the doped and undoped films but the authors do not comment this.

Response 17: The authors thank the reviewer for the comments and the suggested references. In this work, our focus is on the study of the optical and electronic properties of Ir complex doped in different host organic materials, namely CBP, mCP, TAPC, and TCTA, which are used as emitting layers in produced OLED devices. The optical properties of host materials were also examined, as references for the modeling procedure of the Spectroscopic Ellipsometry (SE) spectra and completeness, as well. It is also important to mention that for the analysis we used the Tauc-Lorentz model, which works particularly well for amorphous organic materials.

In addition, SE is a commonly used tool to determine the thickness of thin films and investigate their dielectric function in a non-destructive way. During SE measurements the reflection properties of the whole sample are recorded. A subsequent evaluation by modeling the measured data is necessary to extract the properties of individual layers. This procedure requires some knowledge of the general sample structure: the dielectric function of the substrate should be known with high precision and also the order of layers (e.g. substrate/oxide/film/roughness) is a necessary input parameter [Lehmann et al. SpringerPlus 2014, 3:82]. It is well known that SE is one of the finest techniques to evaluate microstructural changes [SN Applied Sciences (2021) 3:500 | https://doi.org/10.1007/s42452-021-04495-7]. So, the analysis of SE concerns the features of thin polymer films associated with their microstructure and morphology [Adv. Funct. Mater. 2014, 24, 2116–2134, DOI: 10.1002/adfm.201303060].

The optical and electronic properties of thin films are affected significantly by process conditions. Solution deposition of organic small molecules involves the dissolution of the deposited material into an organic solvent where it can then be deposited onto a substrate. As the solvent evaporates the solution becomes supersaturated, driving nucleation and crystal growth, to form a thin film. Compared to vacuum, the nucleation and growth of solution deposited materials are more complex due to added solvent–vapor, solvent–substrate, solute–solvent, and solute–substrate interactions. Additionally, control over the formation of thin films by solution processes is limited due to the rapid progression of nucleation, crystallization, and growth stages that can occur in a matter of seconds [RSC Adv., 2021, 11, 21716–21737, DOI: 10.1039/d1ra03853b]. So, it is interesting to examine not only the quality of film forming properties but also the optical behavior of doped and net hosts via solution deposition methods. According to this reference (see J. Phys. Chem. C 2020, 124, 11701, DOI: 10.1021/acs.jpcc.0c01968), the authors study and compare changes in the PL characteristics of the films based on similar thickness made by vacuum deposition versus solution-coating. However, it is observed that the PL emission profile of the solution fabricated samples presents differences compared to the vacuum deposited films for the doped materials based on CBP: Ir(ppy)3, specifically for the pristine spin coated CBP: Ir(ppy)3 the emission profile blue-shifted compared to the vacuum, and this small shift could be affected to the emission color for the OLED technology. So, film thickness and microstructure are crucial parameters that impact on Photoluminescence, because of its various nonlocalized (aggregate states) and localized (intrachain) transitions manifestations [Polymers 2022, 14, 641. https://doi.org/10.3390/polym14030641].

Indeed, we agree that in Fig. 1 (b) there are differences between the doped and undoped films, which may be related to the microstructure of the thin films. Nevertheless, as it is stated in Lines 202-204 of the manuscript: “It should be noted here that there are only small modifications in the characteristics of the electronic absorptions of all the host organic materials at higher energies.” That is the electronic absorption features are preserved between the undoped and doped films.

A comment at the beginning of the second review can be considered as the answer to that point.

Point 18: Authors of the manuscript calculate absorption coefficients (Fig. 2) but absorption measurement is probably the most common spectroscopical technique which can give the exact spectrum derived from an experiment while simulations can be less trustful. In fact, in figure 2 a, c, d in the manuscript the spectra have tails till 700 nm, while in figure 2 b the absorption coefficient decreases rapidly before that wavelength. Taking into consideration the absorption spectrum of the complex in Adv. Mater. 2011, 23, 2721, DOI: 10.1002/adma.201100405, where the complex does not absorb above 650 nm, the last figure seems to be correct and figures 2 a, c, d not. On the other hand, small doping level of the samples could have little effect on their absorption. All these doubts can be easily resolved with absorption spectra measured experimentally.

Response 18: In this comment, we would like to express our disagreement, since the calculation of the absorption coefficient by means of spectroscopic ellipsometry is incontrovertible and can provide valuable information about the absorption edge, which is also affected by the defects or structural changes and disordering of organic doped films and not only by the optical properties of the dopant.

I can agree with the authors.

Point 19: In line 86 the authors write “According to this analysis, the emphasis will be placed on the energy transfer mechanism between the organic host material and the metal complex”. In the manuscript there is a discussion of energy transfer mechanisms in organic semiconductors but only at a general level. There is no analysis of the overlap of the complex absorption spectrum with the PL spectra of the hosts, no Forster radii are calculated or samples with different doping levels are not investigated, no transient PLs or lifetimes are measured. From the analysis done by the authors no information can be derived which host transfers energy to the dopant the best and why.

Response 19: The energy transfer mechanism that takes place from the hosts to the dopant is evident in Fig. 3 (which shows the PL spectra of net hosts and the Host:Ir-complex films) and it seems to be similar for all Host:Ir-complex films. A more thorough study of the energy transfer mechanism could be the subject of future work.

Probably I was misunderstood here. I agree that Fig. 3 proves that the energy transfer from the hosts to the guest occurs. However, this alone (and there is nothing more) is not enough to state in line 86 “… the EMPHESIS will be placed on the energy transfer mechanism between the organic host material and the metal complex”. There is the proof that energy transfer takes place and that is all, there is no experimental emphasis to understand the transfer mechanism and the discussion is on a general level (due to a lack of other experiments that could give insight into the transfer processes). It is enough to rephrase the sentence.

Point 20: EL measurements with the same complex done by others: Adv. Mater. 2011, 23, 2721, DOI: 10.1002/adma.201100405 Adv. Funct. Mater. 2013, 23, 4105, DOI: 10.1002/adfm.201300187

ACS Photonics 2015, 2, 9, 1366, DOI: 10.1021/acsphotonics.5b00346 NATURE COMMUNICATIONS 5:4769, DOI: 10.1038/ncomms5769

 In all the above articles OLEDs were evaporated while in the manuscript the authors produce organic layers with spincoating. However, a device production method should not have strong influence on the shape of EL spectrum (the emitter is the same).

Response 20: We think that the device production process, such as spin coating is a crucial factor for the film-forming properties of the investigated materials and it is a significant step forward in the development of flexible red OLEDS, for example for the biosensing applications. It is novel the fabrication of thin films based on small molecules via solution deposition methods, as they are low- cost fabrication processes and provide ease of fabricating large-area devices. On the other hand, small anisotropy of molecular orientation exists, and there is difficulty in fabricating an ideal multilayer structure [J. Mater. Chem. C, 2015, 3, 11178—11191]. So, there is a need for the investigation of emitting films based on Host: Ιr complex by spin coating technique and then applied in single layer OLED structure.

In addition, the EL emission depends on the injection, transportation, and recombination of the carriers along the device architecture, so it is important to examine the EL emission profile. Sometimes, there is a probability of self-quenching emission by the phosphorescent dopant in a neat film. So, the implementation of the host material in the emissive layer is indispensable for the fabrication of PhOLEDs. In particular, the selection of host material for the phosphorescent dopant in the emitting layer, fundamentally affects the efficiency of energy transfer from the host to the dopant, and this fact results in defining the emission characteristics of the photoactive layer. Except for this fact, the adoption of host materials is also conducive to facilitating carrier injection and transportation. The hosts are responsible for balancing the charge transport and mainly contributed to the confinement of electron-hole pairs within the emitting layer. Consequently, the choice of host materials is of great importance for EL profile. The EL emission profile is not attributed only to the emitter-dopant, as the host material and the interfaces between the layers play a significant role.

I can agree with the authors, except that the application of small molecules as hosts in solution processed OLEDs is novel, see for example a review Adv. Mater. 2014, 26, 4218 DOI: 10.1002/adma.201306266. This reference (or similar and more recent review) should be included in the introduction.

Point 21: Moreover, all the host molecules investigated by the authors are predominantly hole transporting materials and in order to achieve balanced charge transport they are typically mixed with electron transporting compounds in spincoated emitting layers. Alternatively, electron transporting layer should be placed on the top of emitting layer to obtain recombination zone far from the electrode, which allows to arrive at higher efficiencies (the performance of the OLEDs is not reported by the authors).

Response 21: We appreciate the reviewer’s insightful suggestion. However, such a suggestion is beyond the scope of our paper, which aims to show that there is the opportunity to use this system Host: Ir Complex, as an emitting layer in order to achieve pure red Solution processable OLEDs based on a simple structure. Thus, based on the presented results we will decide which of these emitting layers is the most promising candidate to apply in red PhOLEDs and we will proceed to the improvement of device performance under further optimization of device architecture.

I can accept that explanation.

Point 22: In summary, it is not clear how some of the measurements help to achieve the goal of the investigation, which seems to be pure red EL. The ellipsometry analysis of dielectric functions has no conclusions and it is never mentioned in other sections of the manuscript. The absorption and PL spectra are discussed in more details but these results are known from earlier publications. The energy transfer is described very generally without entering into details of the processes of the particular host:complex pair. Finally, from the comparison of the PL and EL spectra, the shifts of the peaks between the two cases are reported but a discussion of the origins of those shifts is missing.

Response 22: The synergy of spectroscopic ellipsometry, photoluminescence, and Atomic Force Microscopy characterization techniques provided the overall investigation and evaluation of the optical, photophysical, and surface morphology of thin films. The dielectric function and the absorption coefficient of the blend materials based on Host:Ir complex were dominated by electronic transitions that were assigned to respective electronic transitions in Host materials, which were identified by the study of single component layers grown by the spin coating technique as control films. Ongoing, after the characterization, the emitting materials are implemented in single layer OLED devices. Quantitative analysis of the PL and EL emission peaks and widths realized for all the studied materials to evaluate their color stability and selectivity. Our results showed the possibility that the phosphorescent Host:Ir (III) complex applied in solution processed OLED. The fabricated OLEDs emit orange-red light, specifically for all studied cases the dominant electroluminescence emission is characteristic of Ir(dmpq)2(acac), with a maximum of 625 nm. Among the host materials, the Ir(III) complex doped in CBP provides pure red light emission with stability during the device operation with CIE coordinates (0.66, 0.33), which are very close to those of the standard red (0.67, 0.33) demanded by the NTSC. These encouraging results are a feasible approach for enriching phosphorescent emitting molecules for practical applications.

Earlier comments can be considered as an answer to that point.

Point 23: Taking into account a small novelty of the data and all the above flaws I do not recommend the article to publication in Photonics.

Response 23: It is significant to study both the Host and Host: Ir complex as thin films, fabricated by spin coating technique, which is compatible with Roll-to-Roll manufacturing. Up to now, a solution process-based device fabrication is considered to open the way to true low-cost mass production of large-area lighting devices, as it has the advantages of easy fabrication and low-cost process. This inspires the exploration of the solution-processable approach by incorporation of Host:Ir (III) complexes. Initial studies about the solution-processed PhOLEDs were focused on polymer as the host material, as it is well-known that the polymers possess superior solubility in common solvents and good film-forming properties with low surface roughness. However, the intrinsic deficiencies of polymers’ uncertain molecular structure, the requirement for a harsh purification method, and the low energy state limit their implementation of these materials in PhOLEDs. Another promising and novel approach is based on the small molecules as host materials, compared to polymers, they possess easy synthesis, high purity, and stable thermal properties. Another important thing is to achieve pure red-light emission, as there is an emerging trend of using OLEDs in wearable technology and more specifically in non-invasive sensors. So, remains an open issue to investigate materials, which are used as emitting layers to apply in single-layer red OLEDs.

Earlier comments can be considered as an answer to that point.

Round 3

Reviewer 2 Report

  The authors have corrected all indicated mistakes and answered all doubts. As the last improvement I would add the AFM thickness measurements into the Table 1 to confront them with the results obtained from the SE characterization. After this correction I recommend the manuscript to publication.
